# PROSPECT: Labeled Tandem Mass Spectrometry Dataset for Machine Learning in Proteomics

Omar Shouman *        Wassim Gabriel*        Victor Giurcoiu        Vitor Sternlicht

Mathias Wilhelm

Computational Mass Spectrometry
School of Life Sciences
Technical University of Munich (TUM)
Freising, Germany
{firstname.lastname}@tum.de

## Abstract

*Proteomics* is the interdisciplinary field focusing on the large-scale study of proteins. Proteins essentially organize and execute all functions within organisms. Today, the bottom-up analysis approach is the most commonly used workflow, where proteins are digested into peptides and subsequently analyzed using Tandem Mass Spectrometry (MS/MS). MS-based proteomics has transformed various fields in life sciences, such as drug discovery and biomarker identification. Today, proteomics is entering a phase where it is helpful for clinical decision-making. Computational methods are vital in turning large amounts of acquired raw MS data into information and, ultimately, knowledge. Deep learning has proved its success in multiple domains as a robust framework for supervised and unsupervised machine learning problems. In proteomics, scientists are increasingly leveraging the potential of deep learning to predict the properties of peptides based on their sequence to improve their confident identification. However, a reference dataset is missing, covering several proteomics tasks, enabling performance comparison, and evaluating reproducibility and generalization. Here, we present a large labeled proteomics dataset spanning several tasks in the domain to address this challenge. We focus on two common applications: peptide retention time and MS/MS spectrum prediction. We review existing methods and task formulations from a machine learning perspective and recommend suitable evaluation metrics and visualizations. With an accessible dataset, we aim to lower the entry barrier and enable faster development in machine learning for proteomics.

## 1   Introduction

Proteins are large multi-functional bio-molecules that are responsible for sustaining life. They perform a vast array of functions that are not limited to providing structure for cells and enabling body movement. Some proteins act as channels in the cell membrane and are responsible for delivering oxygen to the body through the bloodstream. Others help build new molecules and act as catalysts to regulate almost all chemical reactions in our cells. They also act as a line of defense in our body since antibodies are made of proteins [1]. Finally, proteins play a vital role when developing treatments [2]. A prominent example is the development of the recent mRNA vaccines that were employed to fight the COVID-19 pandemic, where they help trigger an immune response to the virus [3].

---

*Equal contribution.

36th Conference on Neural Information Processing Systems (NeurIPS 2022) Track on Datasets and Benchmarks.

The collection of all proteins expressed at a given time is called a proteome. While genetic information remains mostly unchanged during an individual's life, its proteome is highly dynamic [4]. It depends on several factors, such as age, and constantly reacts to both internal and external influences. These dynamics can therefore provide insights on fundamental biological questions and help understand and treat diseases. [5].

Mass spectrometry (MS) [6] is an essential tool in proteomics for high throughput identification and quantification of proteins. The objective is to measure the mass-to-charge (weight) of molecules and number of molecules (intensity). In essence, we would like to figure out which molecules exist in what abundance. In tandem mass spectrometry, individual peptides are fragmented. A fragmentation spectrum is recorded for each peptide, allowing the deduction of their amino acid composition. Through the analysis of spectra, the identity and quantity of proteins in the original sample can be inferred. These identification and inference procedures are highly computerized, allowing researchers to use them efficiently in large-scale experiments to identify a large number of peptides in a short time frame [7]. Experimentally, we commonly see only up to $50\%$ of the spectra being identified; therefore, there is room to improve the process using predictions from machine learning models.

In recent years, machine learning algorithms successfully improved the identification of peptides, which allows to infer the presence of proteins. Essentially, machine learning models are trained to predict what the measurements in a mass spectrometer would look like for a given peptide. The experimental measurements are then compared with the predicted measurements from the model in order to identify peptides [8]. Both classical machine learning models such as Support Vector Machines (SVMs) [9] and deep learning models [10, 11, 12, 13] were used to predict properties of peptide sequences.

In this paper, We review different approaches and recent methods for formulating two main tasks in proteomics as machine learning problems and introduce a reference annotated proteomics mass spectrometry dataset (PROSPECT [14]). We also recommend evaluation metrics and plots for the two tasks and report results from benchmarking on one recent deep learning model. Our dataset provides a foundation for various proteomics applications and is not limited to the two tasks in focus. With this work, we aim to streamline the efforts in training and evaluating machine learning models, accelerate research, and improve reproducibility and comparison of results.

## 2    Related Work

Deep learning methods outperformed classical machine learning methods and other domain-specific techniques used in proteomics to predict the properties of peptides. Inspired by models from natural language processing (NLP), proteomics researchers developed and trained various model architectures on different datasets. The datasets used for training are usually either measured, curated, and pre-processed internally in each research laboratory or collected from public repositories such as PRIDE [15] and MassIVE [16]. As summarized in Figure 1, we present a high-level overview of a typical machine learning task setup, looking at inputs, common model architectures, and applications in proteomics. This setup does not apply to each and every machine learning problem in proteomics (e.g. De novo sequencing does not follow it), but can be considered a common ground for a variety of tasks. For a listing of different tasks, we refer to Section H in the appendix.

### 2.1    Inputs and Encoding

The primary input in several proteomics applications is the peptide sequence, a variable-length sequence of amino acids represented by alphabetical letters. Sequences go through two essential pre-processing steps before being fed into a deep learning model: encoding and padding. Encoding is required to allow for consuming the sequences by different neural network layers. One-hot encoding is a common technique to numerically represent sequences by replacing each amino acid with a one-hot vector. The resulting encoded representation of a sequence is a 2D matrix with dimensions $N \times L$, where N is the total number of unique amino acids in the data and $L$ is the sequence length. One-hot encoding is used in DeepLC [11]. An alternative encoding technique is to learn representations for each amino acid by having an embedding layer, similar to NLP models. The weights of the embedding layer are jointly trained with the other weights of the model. The size of the embedding matrix is $N \times D$, where N is the total number of unique amino acids in the data and $D$ is a user-specified dimensionality for the embeddings. Note that after looking up the embedding

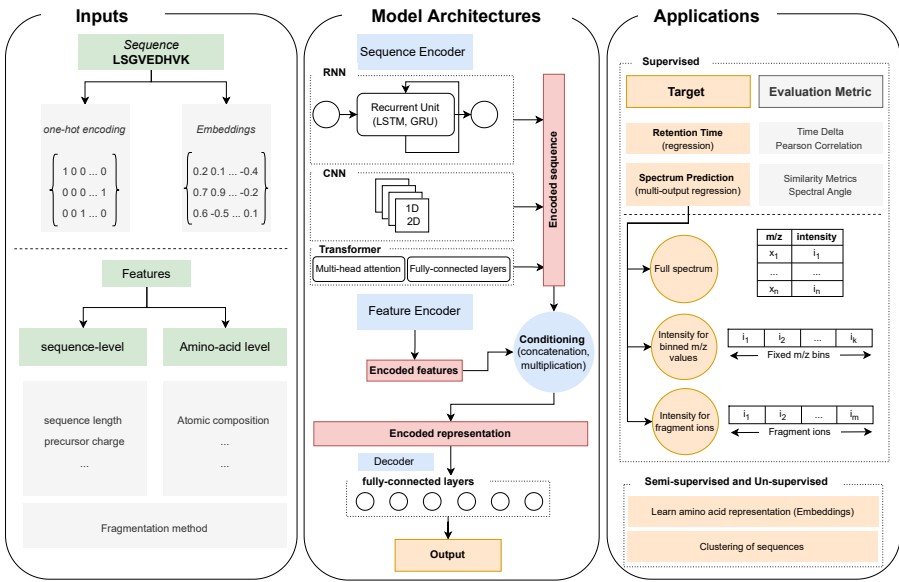

Figure 1: High-level overview of a typical machine learning setup in proteomics for several tasks.

matrix, a sequence representation will be of dimensions $D \times L$. Embeddings are used in DeepRT and Prosit [10, 8, 17]. Padding is mandatory for models that require a fixed length for all sequences. Each sequence is padded, mostly with zeros, up to a certain sequence length $L$, a user-specified parameter. Padding is used in various models, such as in Prosit [10]. For models accepting a maximum fixed sequence length, longer sequences are not truncated but rather ignored during training and prediction.

Several models leverage secondary inputs to provide the model with more context and information about the sequences. In the proteomics literature, some models refer to these inputs as meta-data [18] or global features [11], basically features coming from domain-specific information about the sequences and the amino acids constituting each sequence. Sequence-level features include precursor charge and mass, while features on the level of amino acids include atomic composition. Features extracted from the data generation process can also be included, such as the used fragmentation method. Features are generally numeric, either integer counts or floating-point values; therefore, requiring no further encoding. Collision energy is, for example, a commonly used additional feature [10, 18]. In cases where the model receives categorical features such as the fragmentation method, they can then be one-hot encoded. Several models in the literature, such as [11, 10] utilize additional features. From a dataset viewpoint, some features can be derived given the sequence (e.g., sequence length). In contrast, others require feature extraction and look-ups (e.g., atomic composition) from publicly available resources, such as Unimod [19].

## 2.2 Model Architectures

Due to the sequential nature of the data in proteomics, researchers adopted several model architectures from the NLP domain. Most architectures process the input sequences with an encoder block, whether they are one-hot encoded or represented as embeddings. The block consists of either convolutional layers or recurrent layers. On the one hand, Convolutional Neural Networks (CNNs or ConvNets) excelled in computer vision tasks because they can learn local features invariant to the location in the input, given their local connections and parameter-sharing architecture. Pooling layers usually follow convolutional layers to aggregate and merge the learned features [20]. For processing sequences, convolutions can learn features from neighboring amino acids. DeepLC [11] is one of the models mainly based on CNNs for encoding the sequences as well as some of the features. On the other hand, Recurrent Neural Networks (RNNs) can capture relations in a sequential input due to their architecture with feedback connections [21]. However, they come at the cost of additional difficulties in training (vanishing and exploding gradients) and longer training time [22]. Two RNN variants are commonly used in the proteomics literature; Long-Short-Term-Memory (LSTMs) [22] and Gated Recurrent Units (GRUs) [23]. GRUs are generally faster to train than their equivalent LSTMs. Prosit

[10, 8] encodes the sequences with GRUs, while pDeep [12] uses LSTMs. Several RNN-based architectures employ bidirectional layers to process the sequence in both directions. The Transformer architecture introduced by [24] outperformed several models for sequential data, specifically in natural language applications. PrositTransformer [18] is a recent model that encodes the sequences with a transformer architecture, leveraging positional encoding and the self-attention mechanism.

Models encode the additional features with a simpler encoder architecture depending on their representation (i.e., values and dimensionality). Using conditioning [25], models involving extra features other than the sequences combine both encoded representations (the two latent spaces). The most common conditioning techniques are element-wise multiplication, used in Prosit [10], and concatenation, used in DeepLC [11].

Later in the architecture, a decoder with mostly fully-connected layers processes the combined representation and outputs predictions in supervised learning tasks. Throughout the model, from one layer to the next, the output is passed through a non-linear function (activation function). Rectified linear unit (ReLU) [26] is the most popular activation function or a variant thereof [20]. Prosit [10] and DeepRT [13, 17] use ReLUs, while DeepLC [11] uses Leaky ReLUs [27]. The output layer of the model has almost always a linear activation function, where the regressed values in supervised learning tasks are passed without transformation.

## 2.3 Applications

Trained deep learning models following the aforementioned general overview produced good predictions of peptide sequence properties in a supervised setting. They outperformed classical machine learning methods as well as traditional domain-specific methods such as index-based, modeling-based, and look-up methods [28].

We focus on two main supervised learning tasks in proteomics: retention time prediction and MS/MS spectrum prediction. Retention time prediction is a regression task where the model predicts the retention time value for a peptide sequence.

The second task is to predict the MS/MS spectrum. The spectrum is two-dimensional, with tuples of values $(m/z, I)_i$ for mass-charge ratios and intensities. This task is more complex than retention time prediction and is modeled as a multi-output regression task. As shown in Figure 1, we could group the different methods of formulating this task as a machine learning problem under three main categories. The first category is to predict all $m/z$ values and their corresponding intensities. The second approach is to bin the $m/z$ values into a fixed number of bins, $k$ for example, and predict the corresponding intensities, as in [29]. Lastly, the third one predicts the intensities for a pre-defined set of $m$ fragment ions, such as in [10, 18, 30, 12]. The two latter approaches are simpler formulations than the first one since the positions of the intensity values in the predicted vector are fixed, either for an $m/z$ bin or a fragment ion type. The first approach is difficult due to the precision and range of the measurements, parts-per-million ($10^{-6}$) for $m/z$ and percent ($10^{-2}$) for intensities.

## 2.4 Data

In the proteomics community, there exists several datasets that are used to train and evaluate models for both spectrum and retention time prediction. We observed that ProteomeTools [31] is one of the most commonly used raw datasets in literature, or sometimes a subset of it. Table 1 shows a summary of recent publications for both tasks, highlighting the datasets used, the associated labels, and the maximum supported sequence length. Comparing performance of new models or training approaches becomes challenging given the size of the datasets, the custom in-house data preparation workflow, and the different data splits.

## 3 Dataset

ProteomeTools [31] is the base for our dataset. It contains multimodal liquid chromatography-tandem mass spectrometry analysis for over a million synthetic peptides, representing all canonical human gene products. PROSPECT (PROteometools SPECTrum compendium) is our annotated dataset leveraging the raw data from ProteomeTools. We chose ProteomeTools for three main reasons. First, it is one of the most common datasets used in the proteomics machine learning literature, as seen in

Table 1: Summary of datasets used in literature for training

| Name | Task | Data | Labels | Peptide Length |
|------|------|------|--------|----------------|
| pDeep2 [32] | Spectrum | ProteomeTools + others | b/y ions | dynamic |
| MS$^2$PIP [30] | Spectrum | Multiple datasets | b/y ions | 28 |
| Predfull [29] | Spectrum | ProteomeTools + others | Binned $m/z$ | 30 |
| Prosit Transformer [18] | Spectrum | ProteomeTools | b/y ions | 32 |
| DeepLC [11] | Retention Time | ProteomeTools + others | RT | 60 |
| DeepRT+ [17] | Retention Time | ProteomeTools + others | RT | 50 |
| Prosit [10, 8] | Both | ProteomeTools | RT & b/y ions | 30 |

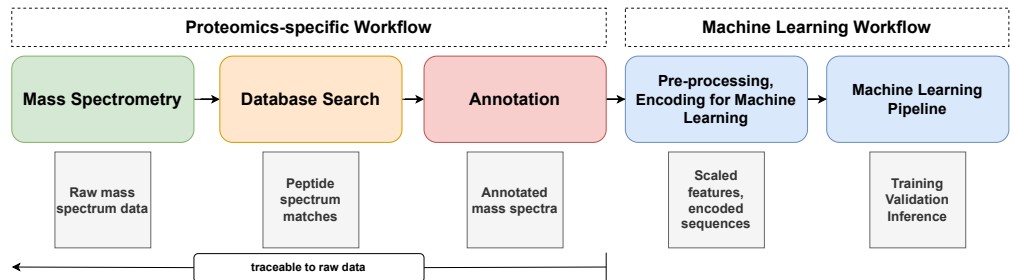

Figure 2: Data workflow and stages in proteomics.

Table 1. Second, it contains high-quality spectra with less noise in the data by reducing co-elution. Third, it contains measurements on synthetic samples of peptides and hence does not contain any reference to human samples. Having only synthetic peptides in the raw data has two advantages. First, we know beforehand which peptides exist in each measurement, achieving a lower rate of false identification in our pipeline. Second, since no human samples are involved, no ethical concerns are associated with the dataset from this perspective.

We refer to the ProteomeTools original publication [31] for more details on its advantages. In the appendix Section G, we also discuss in more details the added value PROSPECT brings on top of ProteomeTools to serve as a reference dataset for machine learning in proteomics.

This section briefly describes the data workflow in proteomics, introduces the dataset schema, and provides some summary statistics and exploratory analyses. Although the dataset enables research in various directions and tasks, we focus on retention time and MS/MS spectrum prediction.

## 3.1 Data Generation and Processing Workflow

The workflow for MS data involves multiple steps before feeding the data to a machine learning pipeline, as depicted in Figure 2. First, samples of peptides are analyzed with the mass spectrometer, which measures the $m/z$ ratio. Through tandem MS, we get spectra for the different peptides. Second, we identify peptides in our samples through a database search. We used MaxQuant [33] in this step, a quantitative proteomics software package. Since our base dataset, ProteomeTools, contains only synthetic peptides, we know which ones to expect, leading to lower false-positive matches. Based on the peptides identified by MaxQuant, we used an expert annotation system [34] to annotate $y$ and $b$ fragment ions (up to triple-charged) as well as possible neutral losses. Providing training data with annotation of neutral losses is up to our knowledge novel, where we annotated 26 different neutral losses and combination thereof, supporting two concurrent neutral losses. Since the variation in the intensity values is high, we scaled the intensities to be in the range between zero and one. Nevertheless, users can always trace the spectra back to the raw data and retrieve the raw intensities. Our implementation of the annotation pipeline is available in a dedicated GitHub repository [35].

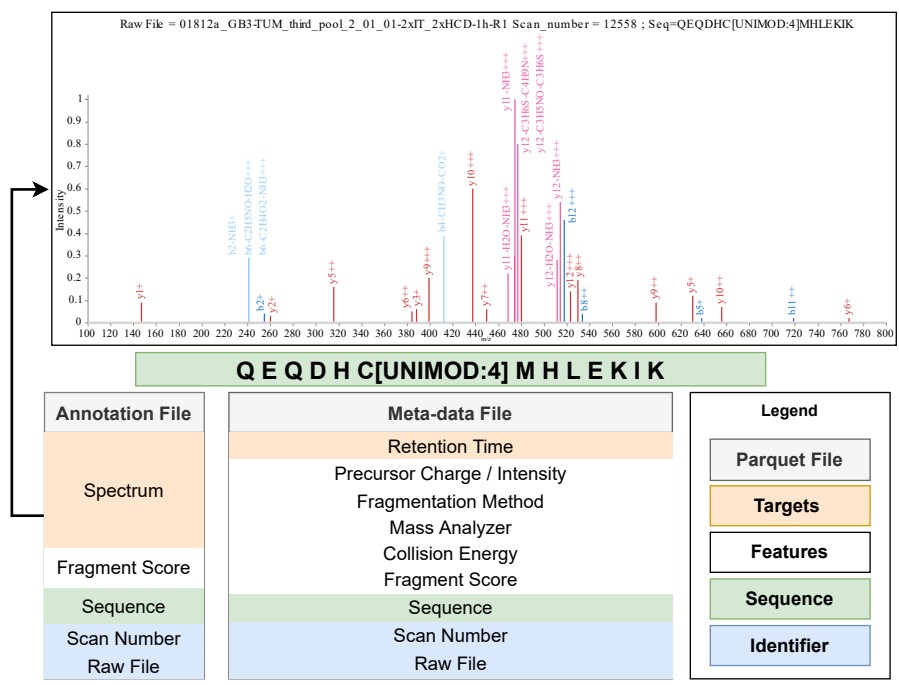

Figure 3: Data schema and a visualized example spectrum with the structure of the data files.

Table 2: Summary statistics of the dataset.

| Packages | Pools | Unique Peptides | Precursors | Spectra | Annotated Peaks | Raw Peaks[1] |
|---|---|---|---|---|---|---|
| 12 | 983 | 838 K | 1.24 M | 61.7 M | 5.7 B | 24 B |

[1] Available in the raw ProteomeTools dataset [36, 37, 38].

## 3.2 Dataset Schema

The dataset consists of 12 packages and has two main parquet file formats; meta-data and annotation files. There is one meta-data file for each package, while the annotations file is split into multiple files per package to facilitate reading the data. Annotation files are sub-organized by pools, where a pool is a set of 1k peptides measured in one analysis run. In both files, a unique identifier to trace back any example to its original raw data file in ProteomeTools is provided. This identifier is the combination of the raw file ID and the scan number. The original ProteomeTools dataset is available on PRIDE [36, 37, 38] and has the same identifier names. We provide a pre-defined data split that is random with one condition; namely, examples for the same peptide sequence should appear in only one of the three splits (training, validation, and test), similar to the logic in [39]. Figure 3 depicts the structure of the files and visualizes one spectrum with its annotations. The files contain sequences, annotations (targets), features, and other meta-data. The complete dataset is hosted on Zenodo [14]. Table 2 shows summary statistics for the data.

The sequences are represented as strings to allow for flexibility of using the dataset in different encoding and machine learning pipelines. Various modifications can occur on the sequences and its constituting amino acids. Two common types of modifications exist in the data; Carboxyamidomethylation which is applied to all cysteine amino acids and oxidation which happens frequently on the methionine. The amino acids are represented by their corresponding letters and the modifications are encoded in Unimod IDs [19], a common database of protein modifications for mass spectrometry. The representation of the sequence follows one of the recommended notations of ProForma [40], a standard notation for writing sequences with modifications. An example sequence is shown in Figure 3. More statistics about the modifications and their occurrences in the dataset are in Section F in the appendix.

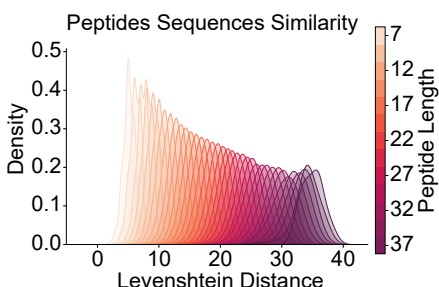 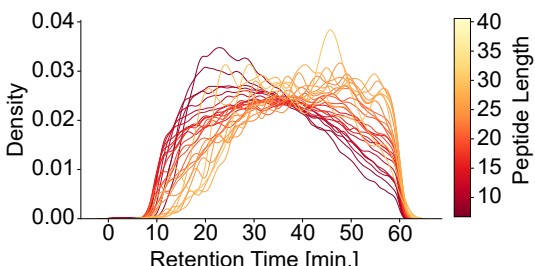

Figure 4: *Left:* Diversity of peptide sequences based on the Levenshtein distance [43]. *Right:* Retention times for different sequence lengths.

Since these two aforementioned modifications are the most frequently observed ones introduced during sample preparation in proteomics, we did not include samples with other modifications to reduce the complexity of the dataset and to first establish a reference benchmarking dataset across the proteomics community. Adding more modifications would be more challenging as it will add another aspect of complexity, where we have to be confident of the position of the modification in the sequence[41, 42].

### 3.3   Exploratory Data Analysis

To illustrate the diversity of the peptide sequences in the dataset, we compute the Levenshtein distance [43] among the different peptides. Figure 4 *(left)* depicts the distribution of Levenshtein distances against the sequence length.

**Data for Retention Time Prediction**

Liquid Chromotography (LC) processes a sample by separating peptides in time according to one of their chemical properties. For instance in Reversed-Phase-LC (RP-LC), which is based on hydrophobicity, peptides with more aliphatic non-polar amino acids such as leucine and isoleucine are more hydrophobic and tend to elute later [44]. The time it takes peptides to elute is denoted as the retention time.

In many cases, models are trained to predict the indexed Retention Time (iRT) instead of the actual experimental retention time. This unitless quantity is calculated by choosing two, or sometimes more, reference peptides as pillars and regressing a line between their retention times. Prosit and DeepLC adopted this approach [10, 8, 11]. In our dataset, we provide the experimental and indexed retention time [45] in the meta-data file. Figure 4 *(right)* visualizes the distribution of retention time values for different sequence lengths in the dataset.

**Data for MS/MS Spectrum Prediction**

Predicting the MS/MS spectrum is a more challenging task. We explained the three main approaches of formulating the prediction as a machine learning task in section 2. Our dataset supports all three approaches. While retention time is determined mainly by the peptide sequence, the spectrum depends on other factors, particularly on the fragmentation method, collision energy, and the charge of the precursor ion.

In the annotations file, all annotated peaks with their intensities and the corresponding $m/z$ values are listed. An example of how a spectrum looks like is depicted in Figure 3. Annotations for $y$ and $b$ ions are included along with neutral losses.

Figure 5 visualizes the distribution of the precursor charge and the average number of peaks per amino acid for different sequence lengths. We can see that the precursor charge tends to be low for shorter peptide sequences and high for longer ones. This distribution shows that the annotations in our dataset cover a wide range of peptides in terms of length and charge states.

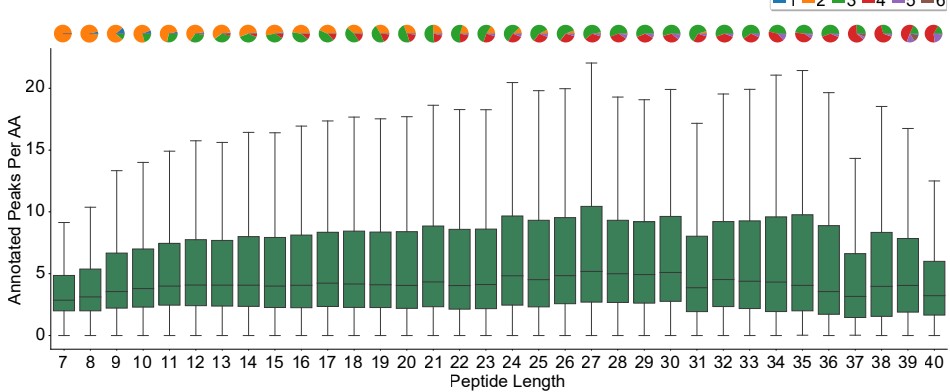

Figure 5: Precursor charge and average number of peaks per amino acid for different sequence lengths.

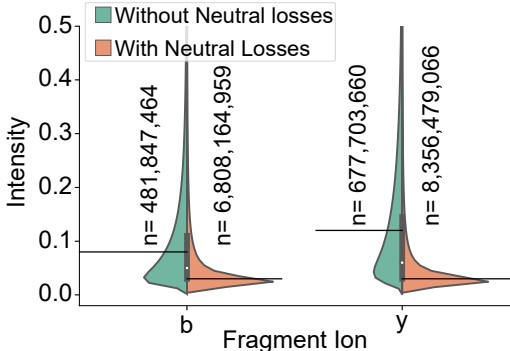

Figure 6: Annotated peaks for $b$ and $y$ ions with and without neutral losses.

The differences between $y$ and $b$ ions with and without neutral losses are visualized in Figure 6. We see that the peaks for neutral losses has lower intensity overall; however, they are useful to explore a different part of the spectrum that is disregarded in traditional search [46].

### 3.4 Impact and Downstream Tasks

Predicting retention times accurately can make MS workflows more efficient by aiding in precursor selection and eliminating ambiguities in peptide identification [47].

Since intensities are usually ignored in traditional database searching [48, 49], being able to predict them quickly and reliably means that valuable new information can be added to traditional pipelines. This re-scoring algorithm improves peptide identification in multiple domains, especially when peptides are very similar. In immunopeptidomics, for example, adding extra features such as peak intensities and retention time helps separate targets and decoys, which significantly boost peptide identifications [8, 10, 50]. Another promising application domain is intelligent mass spectrometric data acquisition, where peptide identifications can help decrease the total time required for experimentation. Peptide features and the validated scoring can be used in real-time to reduce long acquisition cycles and hence increase the efficiency of acquisition [47, 51].

## 4 Experimental Results and Evaluation

One of the main objectives of learning to predict retention times and intensity spectra is to improve peptide identification. One way to evaluate the performance of such models is by integrating them into an application and quantifying their contribution to the overall performance improvement. Such evaluation relies on an extrinsic metric, where we evaluate an improvement in a component to

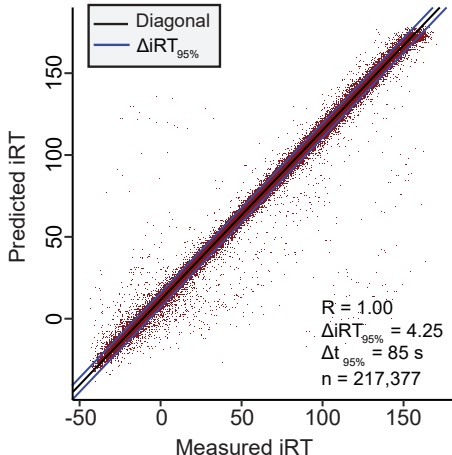

Figure 7: Linear plot with associated metrics for evaluation of retention time models.

know if it will benefit the task at hand [52]. In several cases, having a quick metric to evaluate the model enables faster iteration and improvement of the component at hand. This intrinsic approach helps avoid the costly integration into an end-to-end pipeline and measures the model's performance independent of a larger system or pipeline [52]. The setup is similar to evaluating language models independently with an intrinsic evaluation metric such as perplexity, against evaluating them by embedding them into a speech recognition application [52]. We focus on the intrinsic evaluation of the two tasks mentioned earlier, recommend respective metrics and visualizations, and report results for training Prosit (a recent deep learning model) with our dataset. More experimental details can be found in Section I in the appendix.

## 4.1 Retention Time Prediction

Since retention time models predict a single continuous value, error metrics, such as mean squared error or mean absolute error, provide an indication of the performance and are usually used as loss functions during model training. Another common metric is the Pearson Correlation Coefficient (PCC or R) [53] between the predicted and the observed retention times. Given that models perform relatively well on predicting single values of retention times (values of PCC close to 0.99) [10, 11, 17], the time delta metric provides more fine-grained and domain-specific insights into the model performance [17, 10, 11]. The time delta at 95% $\Delta t_{95\%}$ is the minimal time window containing the errors (residuals) between observed and predicted retention times for 95% of the peptides [13]. The 95% threshold corresponds to $2\sigma$ of the residuals. This threshold can be increased to a higher percentage for stricter evaluation of model performance. A high correlation value is not always guaranteeing good performance in down-stream tasks. The time delta 95 with its domain-specific interpretation can give a clear indication on the time window for identification of eluting peptides. Figure 7 visualizes the linear relationship between measured and predicted iRT values from training and evaluating a Prosit [10] model on our dataset. We recommend a similar visualization for assessing model performance, along with $R$ and $\Delta t_{95\%}$. We provide code to compute the metrics in our data GitHub repository [54].

## 4.2 MS/MS Spectrum Prediction

Evaluation of models predicting MS/MS spectra involves using a distance or a similarity metric between the predicted and the observed intensity vectors. We recommend the normalized spectral angle as an evaluation metric [10] and provide code to compute it in our dataset GitHub repository [54]. It is defined as follows for $V_a$ and $V_b$ being the observed and predicted intensity vectors:

$$SA = 1 - \frac{2}{\pi} \cos^{-1}\left(\frac{V_a \cdot V_b}{\|V_a\| \cdot \|V_b\|}\right)$$

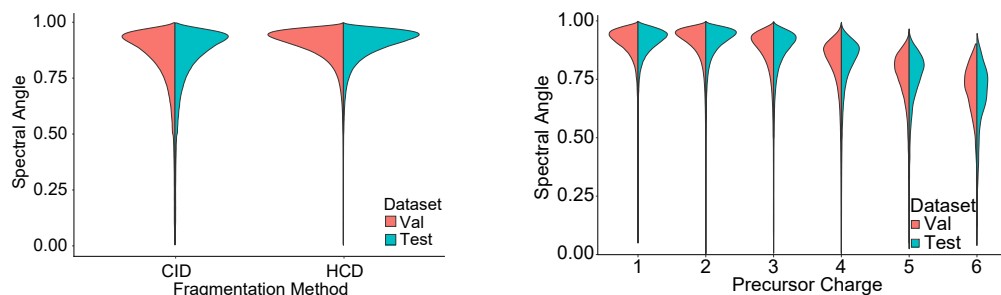

Figure 8: Spectral angle between predicted and observed spectra for validation and test splits.

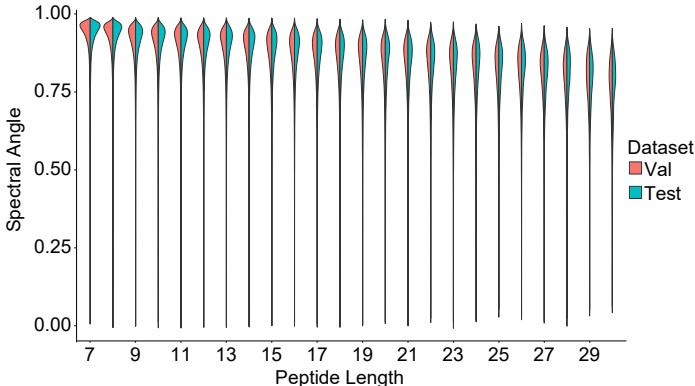

Figure 9: Spectral angle for different peptide length for validation and test splits.

A typical evaluation approach is to see how models perform for different subgroups in the data. For example, models usually perform worse on longer sequences, which can be observed by plotting the spectral angle against sequence length. Similarly, we can evaluate performance for different fragmentation methods (e.g. Collision Induced Dissociation and Higher-energy Collisional Dissociation, CID and HCD) or for different precursor charges. Figure 8 shows our recommended violin plot for evaluation of different fragmentation methods and precursor charges. Figure 9 shows the spectral angle for different peptide lengths. The results are from training and evaluating a Prosit model [10] on our dataset with the recommended split.

## 5   Conclusion and Limitations

In this work, we introduced PROSPECT, an annotated dataset for MS proteomics research based on ProteomeTools [31]. Although the dataset is not limited to retention time and MS/MS spectrum prediction, we focused on the two tasks and recommended metrics and visualizations for model evaluation. PROSPECT provides an annotated dataset for both tasks, allowing for multiple extensions with respect to MS/MS spectra prediction. Along with common annotations for fragment ions ($y$ and $b$), it includes annotation for neutral losses and multiple charge states. The dataset does not include any reference to human samples since it is based on synthetic samples from ProteomeTools [31].

Our dataset represents a starting point to align the efforts in proteomics research. Since the dataset includes a limited number of modifications on peptides, there is potential for extending it with more examples of modified peptides. Similarly, we focused on two supervised learning tasks; however, the dataset enables researchers to formulate and explore new tasks. Examples of supervised tasks include fragment presence prediction [55], de novo sequencing [56, 57] and prediction of the elution profile for retention time instead of one single value. Additionally, the data can be used for self-supervised learning tasks such as learning embeddings of peptide sequences and spectra, [58, 59] and unsupervised tasks, such as clustering of spectra [60]. Detailed listing of supported tasks is in Section H in the appendix.

## Acknowledgments and Disclosure of Funding

This work was in part funded by the German Federal Ministry of Education and Research (BMBF; Grant No 031L0008A) and European Union's Horizon 2020 Program under Grant Agreement 823839 (H2020-INFRAIA-2018-1; EPIC-XS).

We are grateful to Zenodo [62] for their support and cooperation in hosting our large dataset.

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
