# OpenReview forum: "PROSPECT: Labeled Tandem Mass Spectrometry Dataset for Machine Learning in Proteomics"
_NeurIPS.cc/2022/Track/Datasets_and_Benchmarks — NeurIPS 2022 Datasets and Benchmarks _

### Official Review · Reviewer_Qgqc · 2022-07-25
**A benchmark for proteomic profiling with room for improvement**

**Rating:** 7
**Confidence:** 4
**Clarity:** Yes, the paper is well written and cl…

**Strengths:**

- The application of machine learning methods for predicting proteomic properties using protein sequence data is well defined. Different stages of the process from data encoding to commonly used architectures are clearly described.
- The authors clearly define two relevant machine learning tasks that this dataset could be used for. This includes the experimental procedure for generating the data as well as the annotations. The evaluation metrics are well suited for their specific tasks.


**Weaknesses:**

- **Specific innovation compared to Proteomic Tools not well described**: ProteomeTools is repeatedly cited as a ‘base’ for the new PROSPECT dataset. However, the authors do not clearly describe in what ways PROSPECT is different from ProteomeTools making their original contribution unclear. They use the term ‘neutral loss’ as one of their important contributions but don’t define in the text what this refers to. What other annotation was newly added to the data from ProteomeTools and how much already exists? This should be listed clearly.
- **Extent of usability of PROSPECT**: The authors claim that PROSPECT would be beneficial for a number of different proteomic tasks but only go into details for two of them. The authors should list what these other tasks are, what kind of annotation or metadata they require and how PROSPECT is able to facilitate this new capability when ProteomeTools is not. Without this information, it is difficult to asses the full impact of this dataset
- **Evaluation of models**: It would be useful if the authors had trained existing methods in the literature on the two selected tasks and compared performance using the defined metrics. This would illustrate the utility of the metrics and also the difficulty of the task: is it already solved or is there room for improvement in existing methods using this dataset? How to existing approaches differ in their performance?
- **Significance of PROSPECT**: The paper focuses on two proteomic profiling tasks that are quite specific and also not well motivated. Are these tasks foundational in studying the proteome? Or do they comprehensively characterize it? If not, it should be clearly specified which other tasks in proteomics could benefit from PROSPECT and what their significance is. Also, further details on how exactly predicting proteomic properties could impact important applications such as the design of therapeutics would strengthen their paper.

Overall, this seems like a promising direction for a new dataset but the authors need to spend more time fully developing this resource and outlining its use and significance.

**Additional Feedback:**

-

**Correctness:**

The evaluation metrics are defined correctly but no model evaluation is performed on this new benchmark.

**Documentation:**

Yes

**Ethics:**

No ethical concerns since data is synthetic.

**Relation To Prior Work:**

No, the authors do not describe sufficiently how their work differs from existing public repositories such as Proteome Tools, PRIDE and MASSIVE. They claim to have added additional annotation to data from Proteome Tools but don't explicitly list what this additional annotation is and how their dataset is an improvement over existing datasets.

**Summary And Contributions:**

This paper presents a new annotated dataset for predicting proteomic properties from sequence data. This task is useful for determining which amino acid molecules exist in a sample and to what abundance. The authors describe existing machine learning models and datasets that are used for this task. They provide a new standardized dataset for predicting multiple proteomic properties and present evaluation metrics for two specific tasks: peptide retention time and MS/MS spectrum prediction.

---

### Official Review · Reviewer_ywz8 · 2022-07-25
**Great paper**

**Rating:** 8
**Confidence:** 4
**Correctness:** Yes
**Clarity:** Yes

**Strengths:**

- Dataset is a novel type of data modality, important for drug discovery, no current SOTA modeling or community efforts for proteomics
- Well written very clear indicating translational skills between experts in mass spec and deep learning
- data has multiple tasks


**Weaknesses:**

- More detail on what retention time is scientifically
- Make it more clear the size and scale of the datasets (in terms of things deep learning practitioners care about #num samples, feature sizes, etc).
- Provide a few more examples of what applications and impacts a model which does well on this task will have. For example, the models you trained have a certain performance, what can you do with them and what impact will this have on the scientific community? What about if your models were 50% better? What would that impact do? this is a good way to motivate not just a benchmark but why this benchmark really matters (think if on CASP we were able to do 50% better than AF2, what impact would this have immediately on drug discovery research).


**Additional Feedback:**

None

**Documentation:**

Yes

**Relation To Prior Work:**

Yes

**Summary And Contributions:**

This paper presents a new deep-learning ready dataset for looking at protein mass spec data. Mass spec is a central instrument for much work across drug discovery and chemistry/biochemistry at large. In particular, proteomics is a growing area of research (esp. for pharmaceuticals, peptides). This dataset is curated and ready for benchmarking where they aggregate and align the data to  a deep learning community.

---

### Official Review · Reviewer_whEf · 2022-07-26
**An annotated dataset of peptide spectrum worth sharing**

**Rating:** 9
**Confidence:** 3

**Strengths:**

The dataset is centered in the most commonly used dataset, which is nice for consistency with previous work. Annotating it supposes a great novelty because it could help homogenize the training strategies and comparison between models, as well as facilitating working with this kind of data. The dataset creation, availability, and maintenance are clearly detailed, which will facilitate the implementation of these benchmark to other models. The dataset is readily accessible as well as code to use it. The exploratory data analysis provides insights into what the dataset looks like and it is very clearly demonstrated in the article.

**Weaknesses:**

I can't find significant weaknesses. A suggestion for future work would be to demonstrate the use of such a dataset in use for a machine learning model.

**Additional Feedback:**

Great job!!!

**Clarity:**

The paper is extremely well written. The writing is organized, easy to follow, and clear and to the point. The figures are clear and well-done. The discussion at the initial sections really help contextualize this work in the field and makes it easier to follow the rest of the paper, congrats on this!

**Correctness:**

The dataset used as a basis of the annotated dataset is very appropriate to keep consistency and it is constructed following a detailed reasoning described and justified in the article.

**Documentation:**

The dataset and some wrapper code to use it are available on zenodo and thoroughly detailed and ready to use.

**Ethics:**

The dataset they use employs only synthetic peptides so it does not contain blood from patients.

**Relation To Prior Work:**

There is an entire section dedicated to explaining the state-of-the-art of datasets and machine learning models in this topic. This section is thorough and properly referenced. The tables comparing models and different training filters for input data really help illustrate the need for a dataset such as this one.

**Summary And Contributions:**

This work is relevant in the field of proteomics analysis with Mass Spectrometry (MS). The current dataset, ProteomeTools consists of raw data. Hence, current models to predict the retention time, spectra given a sequence, or both have been processing the data in many different ways. This makes it hard to compare these models.

In this work, the authors annotate the existing dataset (ProteomeTools) and include useful annotations like the identity of the peptide, binarized spectra values, normalized intensity values, retention time and metadata related to the experimental details involved in data acquisition. By organizing the data in separate files and filetypes the authors provide an easy way to browse through the data. The annotated dataset does not only lower the entry difficulty in training such models, but also provides an standardized way to train and compare them.

---

### Official Review · Reviewer_figy · 2022-07-27
**No model validation of tasks proposed**

**Rating:** 3
**Confidence:** 3
**Correctness:** N/A

**Strengths:**

The paper highlights an area of research that is not as commonly represented in CS/ML conferences. They reference several papers broadly interested in MS/MS predictive tasks that employ a variety of different architectures.

**Weaknesses:**


(1) This paper proposes tasks, but does not train any models to verify the utility of their dataset, or the validity of the tasks provided. It is unclear what the benchmarks are, and how the community can define "improvement" is, based on the dataset. It is also unclear how generalizable these tasks are to the broader protein space.

(2) It is unclear how they generated data. Section 3.1 discusses data generation, but I could not find discussion on the experimental details of preparing the peptides/experimental samples.

(3) Their justification for using ProteomeTools is that it is commonly used, has high quality spectra, and only uses synthetic peptides. The former two points raise the question why MassIVE data was not used, which also has the same properties. The latter point on why synthetic peptides is more attractive than human peptides is non-obvious to someone not well-versed in MS/MS. Moreover, it is not clear why this property is relevant in benchmarking models.

(4) Their review focuses on sequence-to-spectra style predictions. However, spectra clustering using sequences as labels, not inputs, has been an established strategy (Frank et al. 10.1021/pr070361e.; Gris et al. 10.1038/nmeth.2343., 10.1038/nmeth.3902., Wang et al. 10.1016/j.cels.2018.08.004.). The authors should include this in their review, as there are several strategies to design these models.

(5) No analysis is done on the sequence diversity/composition of the input sequences. How much variety is there in the sequence similarity of the peptides?

(6) Authors pre-filtered most modifications except for 2 from their dataset, which biases benchmarks. For the two included, what fraction of the dataset contains sequences with these modifications?

(7) The retention time data plot seems to suggest there is substantial overlap between sequences of different lengths that have the same retention time. However, protein models can be quite sensitive to input length. How does this benchmark consider models that may use trivial heuristics, like input length, to perform well on a predictive task?

(8) The authors annotate neutral loss in their dataset, but do not explain why this may improve the quality of benchmarking.

(9) The authors seem to note that the retention time regression task is trivial to compute ( R^2 ~= 0.99) from prior work. If the task is this easy, then why is it considered a benchmark? The authors propose a time-delta, but are unclear on what "domain insights" this could provide. Could the authors cite examples on how this refined resolution is beneficial?

**Additional Feedback:**

The authors could strengthen their work by testing a panel of different architectures on the tasks they have illustrated. Comparing and contrasting these models, the inductive biases within them, and exploring what makes peptide-to-spectra predictions effective would make this a stronger paper.

**Clarity:**

The term "loss" is overloaded in the paper - there is loss from the experimental context of fragmentation from MS/MS, but there is also loss considered when training models. The authors should clarify what they mean by loss when they use the term, as people unfamiliar with the space may not understand its context.





**Documentation:**

The README provided in the github repo provides an example on how to download the dataset, via a python function or through zenodo. Explicit tutorials or scripts on how to customize/modify/evaluate models using this dataset are not provided.

**Relation To Prior Work:**

The paper does review several other sequence-spectra prediction tasks.

**Summary And Contributions:**

This study curates and annotates the ProteomeTools Dataset, a proteomics dataset that can be used to train models in predicting properties for mass spectrometry (MS/MS) using experimental data. The authors propose 2 tasks: peptide retention time and spectrum prediction. The authors also provide an API to download and extract relevant data, and discuss metrics to evaluate the tasks proposed.

---

### Official Review · Reviewer_jyVC · 2022-07-27
**A useful prospect for improving ML in MS analysis**

**Rating:** 8
**Confidence:** 4
**Correctness:** Appears correct.
**Clarity:** Paper is clearly written.

**Strengths:**

--The combination of metadata, overview of the dataset composition, and recommended metrics and visualizations is a strength and will help new comers into the world of proteomics analysis with MS.
--Much of the paper reads like a review paper, which is also helpful for those new to the protein MS field.

**Weaknesses:**

--The dataset is not new, though has useful additions
--The number of benchmarks presented here are few to none

**Additional Feedback:**

--Figures are nice.

**Documentation:**

Dataset is available on zenodo. It appears well annotated. However, no code repository was found that could be used to reproduce the analyses. It seems that would need to be made available before acceptance?

**Ethics:**

No ethical issues identified.

**Relation To Prior Work:**

Yes, much of the paper reviews work related to this area.

**Summary And Contributions:**

This paper presents a newly annotated and integrated protein MS dataset based on the ProteomeTools dataset. Here, the authors have combined all of this data into a single resource, with associated metadata. The authors then present what they consider to be a sort of "best practices" for evaluating and presenting analyses on two different predicitions tasks (retention time and MS/MS spectrum. The ultimate goal here is to provide an efficient resource dataset for training new ML models that can be used to make protein MS technology more accurate and sensitive.

---

### Official Review · Reviewer_bbps · 2022-07-28
**A good proteomics dataset, but insufficiently original for inclusion in this track**

**Rating:** 3
**Confidence:** 4

**Strengths:**

The application area is important, and the authors produce a package which could make the use of ProteomeTools data more readily available to the ML community. The figures in the paper give a good summary of the database.

**Weaknesses:**

The primary weakness of the paper is the relatively limited original content. The package is a fairly straightforward extension of the ProteomeTools database, which has been used in many studies of peptide LC-MS/MS previously. In my opinion, it is not comparable to papers that have been accepted to this NeurIPS track in the past.

More detailed comments:

1) There is little guidance in the paper or package on how to do data splits. Furthermore, the paper does not provide baseline methods for benchmarking.
2) One of the tasks included in the package is Retention Time prediction. Section 4.1 makes it clear that existing methods already produce very accurate predictions, so it isn't clear why this is a task worth highlighting.
3) The authors acknowledge that extrinsic validation metrics, based on downstream tasks, such as fragment presence prediction or de novo sequencing, are more meaningful. However, the package only suggests a few intrinsic validation metrics for Retention Time prediction and Spectrum prediction. The paper would have been stronger if the authors had included datasets for downstream tasks and some clearly defined benchmarks.
4) The prediction method used to produce Figure 8 is not included.
5) No evaluation metric is provided for spectrum prediction when the peak identities are also predicted.
6) Further discussion on the meaning of y and b fragment ions and neutral losses would be helpful for non-experts. A description of the expert annotation system for losses used (reference [34]) would also be good.
7) In Section 3.2, it is not clear what this sentence means: “to keep complexity low and identification rate high”.


**Additional Feedback:**

The paper would be much stronger if there were meaningful downstream tasks and baseline prediction methods included.

**Clarity:**

The paper is fairly clear, though there is room for improvement (see my detailed comments above).

**Correctness:**

The dataset is constructed in a sound way. There isn't enough benchmarking in the paper, nor a thorough description of what is done.

**Documentation:**

The package is hosted on Zenodo. I was not able to find a maintenance plan.

**Ethics:**

Not as far as I can see.

**Relation To Prior Work:**

The authors could make it clearer how their contribution goes beyond the dataset available from ProteomeTools.

**Summary And Contributions:**

The authors introduce PROSPECT, a database for proteomics research based on a massive collection of LC-MS/MS datasets already available from the ProteomeTools project. The paper is well written, and the package adds some functionality to the original database. In particular, there are some advantages over existing packages in terms of usability, and the authors provide peak annotations from an expert annotation system which is publicly available (reference [34] in the manuscript).

---

### Meta-Review · Area_Chair_wWr3 · 2022-09-10

**Recommendation:** Accept
**Confidence:** 3

**Metareview:**

Reviews were split. Reviewers agreed on the importance of the domain but differed in how they valued the contribution over ProteomeTools, which the dataset builds upon. A primary concern was lack of baselines/benchmarking. The rebuttal argues that baselines would not add much value since strong methods have been published in prior work. However, my interpretation of the reviewers ask is not to invent new heuristic baselines but to evaluate the prior methods on the newly prepared dataset. The authors do add evaluation of one such model (Prosit) to the revision. Evaluating more models in the same way could strengthen the contribution, and could be part of the ongoing maintenance of the dataset / code.

Overall, however, the majority of reviewers felt the paper does provide significant value beyond ProteomeTools and other prior work, through the new annotations, tooling, and usage recommendations. The revision adds an appendix (G) to clarify these contributions, and it may be useful to move some of that text into the intro of the paper (as I think many readers will want to know what are the main contributions beyond the base dataset). I agree with the majority that there is significant value added and recommend acceptance.

---

### Decision · Program_Chairs · 2022-09-16

Accept